

# Estimating global gross primary productivity using chlorophyll fluorescence and a data assimilation system with the BETHY-SCOPE model

Alexander J Norton[1], Peter J Rayner[1], Ernest N Koffi[2], Marko Scholze[3], Jeremy D Silver[1], and Ying-Ping Wang[4]

[1]School of Earth Sciences, University of Melbourne, Melbourne, Australia
[2]European Commission Joint Research Centre, Ispra, Italy
[3]Department of Physical Geography and Ecosystem Science, Lund University, Lund, Sweden
[4]CSIRO Oceans and Atmospheres, Aspendale, Australia

*Correspondence to:* A.J. Norton (alexander.norton@climate-energy-college.org)

**Abstract.**

This paper presents the assimilation of solar-induced chlorophyll fluorescence (SIF) into a terrestrial biosphere model to estimate the gross uptake of carbon through photosynthesis (GPP). We use the BETHY-SCOPE model to simulate both GPP and SIF in a process-based manner, going beyond a simple linear scaling between the two. We then use satellite SIF data from the Orbiting Carbon Observatory-2 (OCO-2) for 2015 in the data assimilation system to constrain model GPP. The assimilation results in considerable improvement between model and observed SIF, despite difficulties in simulating large SIF values due partly to uncertainties in the prescribed LAI. SIF-optimized global GPP increases by 7% to $137 \pm 6$ $\mathrm{PgCyr}^{-1}$ and shows improvement in its global distribution relative to independent estimates. This change in global GPP is driven by an overall decline in APAR and increase in the light-use efficiency of photosynthesis across almost all ecosystems. This process-based data assimilation opens up new pathways to the effective utilization of satellite SIF data that will improve our understanding of the global carbon cycle.

## 1 Introduction

Through photosynthesis terrestrial plants fix atmospheric carbon dioxide ($CO_2$) into organic compounds constituting the largest carbon flux on Earth. This process is the first step in terrestrial carbon sequestration and plays a critical role in offsetting anthropogenic carbon emissions (Campbell et al., 2017; Janssens et al., 2003). However, gross uptake of $CO_2$ through photosynthesis (GPP; Gross Primary Production) cannot be observed at large spatial scales, which limits our understanding of its spatiotemporal distribution and response to climate (Schimel et al., 2015). Furthermore, it limits our ability predict the terrestrial net $CO_2$ flux under future climate conditions (Friedlingstein et al., 2014; Sitch et al., 2015).

Numerous approaches have been developed to estimate GPP at the necessary spatial scale (see Anav et al., 2015). One approach takes existing observations and merges them with process-based models using model-data fusion ('data assimilation') techniques. Data assimilation provides a way toward improving models and evaluating their consistency with various observa-



tional data (Rayner, 2010). The Carbon Cycle Data Assimilation System (CCDAS) is one example that has been developed to ingest multiple sources of data (Kaminski et al., 2013; Koffi et al., 2012; Rayner et al., 2005). The approach typically applies an algorithm that adjusts model state variables, initial conditions and/or process parameters to reduce the mismatch between the observations and their model simulated counterpart whilst accounting for uncertainties. This provides an optimal estimate

of the model variables (e.g. carbon fluxes) conditioned by the observations. By explicitly accounting for uncertainties of the inputs in a probabilistic framework the uncertainty of estimated carbon fluxes before (*a priori*) and after (*a posteriori*) the addition of the observational data can be assessed (Rayner et al., 2005).

A range of observational data have been applied within carbon cycle data assimilation systems (see reviews Kaminski et al., 2013; Macbean et al., 2016). This includes atmospheric $CO_2$ concentration (Rayner et al., 2005), FAPAR (Fraction

of Absorbed Photosynthetically Active Radiation) (Kaminski et al., 2012; Kato et al., 2013), NDVI (Normalized Difference Vegetation Index) (Peylin et al., 2016), soil moisture (Scholze et al., 2016), and in situ flux tower measurements of the net carbon flux and latent heat flux (Bacour et al., 2015; Peylin et al., 2016).

Recent remote sensing measurements of solar-induced chlorophyll fluorescence (SIF) (Frankenberg et al., 2011b; Joiner et al., 2011) offer a novel insight into the spatiotemporal patterns of GPP (e.g. Duveiller and Cescatti, 2016; Guan et al., 2015;

Joiner et al., 2014; Li et al., 2018; Luus et al., 2017). Many studies have shown that SIF correlates strongly with GPP across ecosystem types and generally performs better at tracking GPP than traditional reflectance-based vegetation measurements (e.g. NDVI, EVI) (Joiner et al., 2014; Li et al., 2018; Luus et al., 2017; Walther et al., 2016; Yang et al., 2015).

SIF and GPP are linked at the cellular level through the light reactions of photosynthesis. To initiate the light reactions of photosynthesis, pigment-protein complexes forming so-called photosystems absorb sunlight energy and convert it into the

chemical energy required to power photosynthetic $CO_2$ fixation. This absorbed energy, or excitation energy, has one of three fates. Firstly, excitation energy may be used to drive photosynthetic electron transport, ultimately powering photosynthetic $CO_2$ fixation (Krall and Edwards, 1992), termed photochemical quenching (PQ). Secondly, excitation energy may be dissipated as heat via a range of mechanisms used to protect photosystems against excessive light-induced damage (Demmig-Adams and Adams III, 2006) collectively termed non-photochemical quenching (NPQ). Finally, excitation energy may be passively emitted

from the chlorophyll pigments as chlorophyll fluorescence. During photosynthesis PQ and NPQ are actively regulated by plants to balance energy supply and demand under changing environmental conditions (Porcar-Castell et al., 2014). Chlorophyll fluorescence therefore responds dynamically to changes in the rates of PQ and NPQ, providing a highly useful non-invasive means of monitoring leaf physiological processes (for reviews see Baker, 2008; Govindjee, 1995; Porcar-Castell et al., 2014). Measurements of artificially-induced chlorophyll fluorescence at the leaf level have been used for this purpose for several

decades (Govindjee, 1995).

With the use of satellite-based instruments global maps of SIF have been produced (Frankenberg et al., 2011a; Guanter et al., 2012; Joiner et al., 2011). Parazoo et al. (2014) and MacBean et al. (2018) have used this data to optimise model estimates of GPP. Parazoo et al. (2014) developed a framework to use SIF alongside model estimates to redistribute global GPP patterns by applying linear scaling between SIF and GPP. They did not, however, optimise model parameters so did not

improve model predictive capabilities. MacBean et al. (2018) did optimise model parameters of a single model (ORCHIDEE).



Following empirical evidence, SIF was related to model GPP using a biome-specific linear scaling. In both cases SIF added useful information and induced large shifts in global GPP. However, SIF was not explicitly modelled and therefore was not compared with the observed SIF to assess performance against the data. Koffi et al. (2015) was the first to combine a process-based model of SIF with a terrestrial biosphere model. Koffi et al. (2015) performed global simulations of SIF and a set of sensitivity tests, demonstrating that the model is capable of utilising the SIF data. Norton et al. (2018) extended the model of Koffi et al. (2015) to include a module for prognostic leaf growth. Using this model they quantified how effectively SIF could constrain uncertainties in model parameters and GPP, finding a reduction in uncertainty of global annual GPP of approximately 73%, a result consistent with the model used in MacBean et al. (2018). However, no formal optimisation algorithm was applied.

This study aims to integrate satellite observations of SIF into a data assimilation system to optimize model parameters and estimate spatiotemporal patterns of GPP globally, furthering the work of Norton et al. (2018). This makes an advance on recent approaches by simulating SIF explicitly using a process-based model. We apply SIF in this process-based data assimilation system, assess its performance against the data, and investigate the SIF-optimized GPP patterns.

## 2 Methods

Here we outline the steps taken to assimilate SIF into the terrestrial biosphere model BETHY-SCOPE. First, we briefly describe the BETHY-SCOPE model. This includes an observation operator that can simulate SIF and thus provides a means of mapping model variables into the observational space. Second, we outline the quantities that are optimized within the data assimilation system. In this study these quantities are BETHY-SCOPE parameters. Third, we describe the satellite SIF observations used. Fourth, we outline the algorithm used to optimize the model process-parameters and the method for error propagation. Finally we give a brief description of the specifics of the experimental setup.

### 2.1 BETHY-SCOPE

BETHY-SCOPE is an integration of the existing models BETHY (Biosphere Energy Transfer Hydrology) (Rayner et al., 2005; Knorr et al., 2010) and SCOPE (Soil Canopy Observation, Photosynthesis and Energy fluxes) (Van der Tol et al., 2009) and builds upon the developments by Koffi et al. (2015) and Norton et al. (2018). The coupling of BETHY and SCOPE enables spatially explicit, plant functional type (PFT) dependent, global simulations of GPP and SIF.

BETHY is a process-based terrestrial biosphere model at the core of the Carbon Cycle Data Assimilation System (CCDAS) (Rayner et al., 2005; Scholze et al., 2007). Full model description details can be found elsewhere (e.g. Rayner et al., 2005; Scholze et al., 2007; Knorr et al., 2010). Briefly, BETHY simulates carbon assimilation and plant and soil respiration within a full energy and water balance. Although we prescribe leaf area index (LAI) to the model, this version of BETHY has an optional leaf area dynamics module for prognostic LAI as described in Knorr et al. (2010). The full BETHY model consists of four key modules: (i) energy and water balance; (ii) photosynthesis; (iii) leaf growth and; (iv) carbon balance. It represents variability in physiology and ecology of plant classes by 13 PFTs (see Table 1) originally based on classifications by Wilson



and Henderson-Sellers (1985). Each model grid cell may consist of up to three PFTs as defined by their grid cell fractional coverage.

**Table 1.** PFTs defined in BETHY and their abbreviations.

| PFT # | PFT Name | Abbreviation |
|---|---|---|
| 1 | Tropical broadleaved evergreen tree | TrEv |
| 2 | Tropical broadleaved deciduous tree | TrDec |
| 3 | Temperate broadleaved evergreen tree | TmpEv |
| 4 | Temperate broadleaved deciduous tree | TmpDec |
| 5 | Evergreen coniferous tree | EvCn |
| 6 | Deciduous coniferous tree | DecCn |
| 7 | Evergreen shrub | EvShr |
| 8 | Deciduous shrub | DecShr |
| 9 | C3 grass | C3Gr |
| 10 | C4 grass | C4Gr |
| 11 | Tundra vegetation | Tund |
| 12 | Swamp vegetation | Wetl |
| 13 | Crops | Crop |

SCOPE (version 1.53) is a vertically-integrated (1D) radiative transfer and energy balance model with modules for photosynthesis and chlorophyll fluorescence (Van der Tol et al., 2009). It utilizes the canopy radiative transfer scheme of SAIL (from Scattering by Arbitrarily Inclined Leaves) (Verhoef, 1984) and the leaf radiative transfer model of PROSPECT that is based upon the optical properties of leaves (Jacquemoud and Baret, 1990). SCOPE incorporates current understanding of chlorophyll fluorescence processes including canopy radiative transfer, re-absorption of fluorescence within the canopy, and the non-linear relationship between chlorophyll fluorescence quantum yield (the ratio of quanta emitted to quanta absorbed) and other quenching processes (Van der Tol et al., 2009, 2014). Leaf level chlorophyll fluorescence is coupled to models for photosynthesis of C3 (Collatz et al., 1991) and C4 (Collatz et al., 1992) vegetation, as well as the Ball-Berry model for stomatal conductance (Ball et al., 1987). A current limitation of SCOPE is that the water balance and horizontal heterogeneity of PFTs are not modelled.

The canopy radiative transfer and photosynthesis schemes of BETHY have been replaced by the corresponding schemes in SCOPE, including the components required for calculation of chlorophyll fluorescence at the leaf and canopy scales. The spatial distribution, vegetation (PFT) characteristics and carbon balance are handled by BETHY. SCOPE therefore takes climate forcing (meteorological and radiation data) and spatial information from BETHY and returns GPP. The coupled BETHY-SCOPE model enables process-based global simulations of GPP and SIF.



## 2.2 BETHY-SCOPE Parameters

In this data assimilation system, the model process parameters to be optimized are the unknown quantities related to SIF and GPP. Parameters can be either global or spatially differentiated by PFT. PFT-dependent parameters enable differentiation between physiological, leaf growth and structural traits. Some key parameters for this study such as the maximum carboxylation

rate at 25°C ($V_{cmax}$) (see Table A1) and chlorophyll a/b content ($C_{ab}$) are considered PFT-dependent. The $V_{cmax}$ parameter is used in most process-based terrestrial biosphere models as it is a parameter of the leaf-scale photosynthesis model of Farquhar et al. (1980). The $C_{ab}$ parameter is a parameter specific to the SCOPE model and an important component of the canopy radiative transfer scheme as it strongly influences both SIF and FAPAR. Parameters for vegetation height and leaf-angle distribution are separated into three PFT classes (see Table A1).

There are 42 parameters of BETHY-SCOPE that are optimized by the data assimilation system (see Table A1). The errors associated with each of these parameters is represented by a Gaussian probability density function (PDF). The mean and standard deviation for the prior parameters are shown in Table A1. Choice of the prior mean and uncertainty follow those used in previous studies (Kaminski et al., 2012; Knorr et al., 2010; Koffi et al., 2015). For new parameters that are not well characterized (e.g. SCOPE parameters) we assign relatively large prior uncertainties and mean values in line with the default

SCOPE parameters and with Koffi et al. (2015) and Norton et al. (2018).

Parameters exposed to the data assimilation system are chosen based on previous sensitivity tests such as those performed by Verrelst et al. (2015) and Norton et al. (2018). This includes leaf composition parameters such as $C_{ab}$, leaf dry matter content ($C_{dm}$), and leaf senescent material fraction ($C_s$). Also included are canopy structural parameters such as leaf distribution function parameters ($LIDFa$, $LIDFb$), vegetation height ($hc$) and leaf width, the prior values for these were obtained from

literature values and are assigned to groups of PFTs that we assume have a generally similar structural form (see Table A1). Photosynthetic parameters are also incorporated, including $V_{cmax}$ and Michaelis–Menten constants of Rubisco for $CO_2$ ($K_C$) and $O_2$ ($K_O$).

Given the uncertainty of the photosynthetic kinetic parameters for dark respiration ($R_d$) and the maximum oxygenation rate ($V_{omax}$) they may also be important to consider particularly for modelling GPP (von Caemmerer, 2000). We therefore include

these as exposed parameters, given as their respective ratios with $V_{cmax}$, $a_{R_d,V_c}$ and $a_{V_o,V_c}$. Given these kinetic parameters affect the relatively specificity of Rubisco ($S_{c/o}$) we calculate $S_{c/o}$ explicitly following von Caemmerer (2000) which differs from the original SCOPE model.

## 2.3 Satellite SIF Observations

We use satellite SIF observations from NASAs Orbiting Carbon Observatory-2 (OCO-2) (Sun et al., 2018). Launched in July

2014 OCO-2 operates in a sun-synchronous orbit with an overpass at approximately 1:30 p.m. local time. It has a repeat cycle of 16 days. Collecting approximately 24 spectra per second it has relatively high data density within the field of view. OCO-2 has a ground-pixel size of 1.3 × 2.25 km² and a total swath width of 10.6 km. Full spatial mapping of SIF is therefore not possible





with OCO-2. However, the high spectral resolution of OCO-2 allows for robust and accurate SIF retrievals (Frankenberg et al., 2014; Sun et al., 2018).

Alternative satellite SIF datasets are also available, including from the GOME-2 and GOSAT instruments (Frankenberg et al., 2011a; Guanter et al., 2012; Joiner et al., 2011). There are benefits and pitfalls in using these alternative data. For example,

GOME-2 and GOSAT provide longer time series going back to 2007 and 2009, respectively. GOME-2 also provides better spatial mapping compared with OCO-2. However, there are known issues of sensor degradation with GOME-2 (Zhang et al., 2018). The advantage of the OCO-2 satellite is that it collects eight times more spectra and has a higher spectral resolution providing more robust and data dense observations. We note that a formal comparison of these other datasets is outside the scope of this study.

We use the OCO-2 processed SIF-lite data files. For details on the retrieval algorithm for the SIF data see Frankenberg et al. (2014). This data is gridded at $2° \times 2°$ spatial resolution, equivalent to the model grid resolution. We exclude soundings collected over water as determined by the corresponding IGBP land classification index (Friedl et al., 2010). We use instantaneous SIF at 757 nm and only soundings taken in nadir mode. Data is also available at 771 nm, however the signal at 757 nm is stronger (Magney et al., 2017) thus we only consider that signal. The annual mean OCO-2 SIF for 2015 is shown below in Fig.

15 1.

In order to address issues of spatial coverage and potential sampling bias of ecosystems that differ from those in the model, we assessed the similarity between the sampled IGBP land classification index of the OCO-2 soundings and the BETHY-SCOPE PFTs used here. Close similarities are found in the occurrence of IGBP land classification indexed biomes and the BETHY-SCOPE PFTs. We therefore do not perform any further filtering of the data.

**2.3.1 Observational Uncertainty**

The calculation of observational uncertainties is an important aspect of any data assimilation study as it partly determines posterior probabilities. We note two rather extreme cases in calculating the uncertainty in the satellite observations of SIF. The first is to take the average of the single measurement precision error, considered an overestimate of the uncertainty. Second is to calculate the standard error, where the average of the single measurement precision error is divided by the square root of

the number of observations, as applied in Parazoo et al. (2014). Use of the standard error almost certainly underestimates the uncertainty as it neglects correlated or systematic errors.

Therefore, to determine the measurement error of SIF ($\sigma$) in a given grid cell ($i$), we sum the single measurement precision error ($\sigma_e$) of each sounding within that grid cell and divide by the total number of soundings ($n_i$). Scaling this by one half scales it closer to the standard error but remains a conservative estimate of the actual error.

$$\sigma = \frac{1}{2} \frac{\sum \sigma_e}{n_i} \tag{1}$$



Calculated uncertainties are shown for January and July 2015 in the Supplement Figs. 1 and 2. Statistical tests on the results outlined further below will allow us to test whether these observational uncertainties are consistent with other aspects of this data assimilation process.

## 2.4 Data Assimilation System

We assimilate observed SIF into the BETHY-SCOPE model in order to optimize model process parameters and provide an observational constraint on spatiotemporal patterns of GPP. For this we require a minimization algorithm, cost function, and error propagation method. A variety of techniques are available for optimization of terrestrial biosphere models and reviews are available (Fox et al., 2009; Kaminski et al., 2013; Macbean et al., 2016; Trudinger et al., 2007).

We utilize a probabilistic framework whereby quantities (e.g. observations, model state variables, model process parameters)
are represented by their probability density functions (PDF). These quantities are treated as Gaussian, thus can be described by their mean and standard deviation. For the model parameters the mean is denoted by $x$ and error covariance matrix by $C_x$. We denote the prior parameter vector and covariance matrix by $x_0$ and $C_{x_0}$, respectively, and the posterior parameter vector and covariance matrix by $x_{post}$ and $C_{x_{post}}$, respectively. For the observations the mean is denoted by $d$. The error covariance matrix in observation space, denoted by $C_d$, combines errors in the observations and in their simulated counterpart i.e. model
(Kuppel et al., 2013). Quantification of model error can be performed through an assessment of model-observation residuals following optimization (e.g. Kuppel et al., 2013). We assess potential model errors in this study, however, we do not explicitly account for this error in the propagation of errors onto GPP hence $C_d$ accounts only for errors in the observations. We point out that the uncertainty is embodied in the error covariance matrices and that diagonal elements represent the variance of the quantities while off-diagonal elements represent error correlations between quantities.

### 2.4.1 Assimilation Procedure

The assimilation procedure finds the posterior PDF for the target variables which, in this case, are the model process parameters. We assume Gaussian PDFs so our posterior PDF is described by its mean and standard deviation. The mean is also the maximum posterior estimate which can be found by minimizing a cost function ($J$). The cost function, shown in Eq. 2, quantifies the difference between the model simulated SIF ($M(x_n)$) and SIF observations ($d$) and the departure of parameter
values ($x$) at each iteration ($n$) from the prior estimate ($x_0$). These differences are squared and normalized by the uncertainties in model parameters $C_x$ and observations $C_d$, respectively, allowing for more certain observations to carry more weight. $J$ thus provides a measure of the model-observed mismatch accounting for uncertainties. We consider the optimization to have converged on an optimal solution when the change in the cost function is less than 1% of the change that occurred during the first iteration.

$$J = \sum \left( \frac{(M(x_n) - d)^2}{C_d} + \frac{(x_n - x_0)^2}{C_x} \right) \qquad (2)$$





To find the minimum of $J$ we employ a quasi-Newton method, which is a variational, iterative technique (p. 69 Tarantola, 2005). This algorithm requires a matrix of partial derivatives of the observable with respect to model parameters, called the Jacobian matrix ($H$), calculated using finite differences. $H$ is a representation of the sensitivity of model simulated SIF to each model parameter.

The quasi-Newton algorithm assumes weak non-linearity in the model. This approximation is better than assuming a linear model, but not as useful as having a model adjoint where the entire parameter space can be efficiently examined (Kaminski et al., 2013). With this assumption the model is presumed to be linear about the point where $H$ is calculated. However, to account for non-linearities in the model we recalculate $H$ after each iteration of the algorithm. Given a single 'global' minimum of $J$, this algorithm will converge upon it (Tarantola, 2005). We acknowledge that it is possible that the algorithm will converge

upon a local minimum in $J$.

    For each iteration $n$ of the algorithm the parameter vector ($x_n$) is updated using Eq. 3. This adjusts for non-linearity by performing a forward run of the full non-linear model at each iteration ($M(x_n)$). It takes the form:

$$x_{n+1} = x_n - \mu \left( C_{x_0} + H^T C_d^{-1} H \right) \left( H^T C_d^{-1} (M(x_n) - d) + C_x^{-1} (x_n - x_0) \right) \tag{3}$$

    where $\mu$ is a step-size (set to 0.1) as required in gradient based techniques (Tarantola, 2005). In a case where the parameter

update produces values that are unphysical (e.g. negative chlorophyll content), they are reset to the nearest physical value for the next iteration.

    Alongside $J$ the reduced chi-squared ($\chi_r^2$) statistic is used to assess the match with the observations. Shown in Eq. 4 below, $\chi_r^2$ measures the goodness of fit per observation accounting for observational uncertainties, where $N$ is the total number of observations.

$$\chi_r^2 = \frac{1}{N} \sum \left( \frac{(M(x_n) - d)^2}{C_d} + \frac{(x_n - x_0)^2}{C_x} \right) \tag{4}$$

    The $\chi_r^2$ statistic encapsulates the size of the average mismatch between the simulation and the observations, accounting for the number of degrees of freedom from the unknowns. A value of one means the mismatch is what we expect given the noise level specified in the observational uncertainty; we are neither over-fitting or under-fitting the data (Michalak et al., 2005).

### 2.4.2   Error Estimation

For linear and weakly non-linear problems Gaussian probability densities propagate forward through to Gaussian distributed quantities (Tarantola, 2005), termed linear error propagation. The posterior parameter errors, $C_{x_{post}}$, are estimated using linear error propagation as shown in Eq. 5 as follows:

$$C_{x_{post}}^{-1} = C_{x_0}^{-1} + H^T C_d^{-1} H \tag{5}$$





where $H$ is calculated at the posterior (i.e. $x_{post}$). Rayner et al. (2005) demonstrated how to propagate parameter uncertainties forward through a model onto simulated quantities such as carbon fluxes. Using the Jacobian rule for probabilities, parameter uncertainties in the model parameter covariance matrix ($C_{x_0}$ and $C_{x_{post}}$) can propagate forward onto GPP using Eq. 6. Note that the model Jacobian with respect to GPP ($H_{GPP}$) is also calculated at $x_{post}$. Using Eq. 6 we can determine the

error covariance of GPP ($C_{GPP}$).

$$C_{GPP} = H_{GPP} C_x H_{GPP}^T \qquad (6)$$

With this we can quantify the change in error covariance of GPP by using either $C_{x_0}$ or $C_{x_{post}}$ in Eq. 6 and calculating the difference between the two.

## 2.5 Experimental Setup

In this study BETHY-SCOPE is run for the year 2015. This constitutes the optimization (or calibration) period. We then assess the optimized model performance against independent OCO-2 observations outside of the optimization period from September-December 2014.

The model is run on a $2° \times 2°$ grid resolution. Model SIF is calculated at the equivalent wavelength as OCO-2 SIF (757 nm) and overpass time (1:00 - 2:00 p.m. local time). Climate forcing data is provided in the form of daily meteorology (precipitation,

minimum and maximum temperatures, and incoming solar radiation) obtained from the WATCH/ERA Interim data set (WFDEI Weedon et al., 2014). These are used to derive average diurnal cycles of climate forcing. Atmospheric $CO_2$ concentration is set to the 2015 annual average of 397 ppm. LAI is prescribed to the model using the MODIS improved LAI dataset (Yuan et al., 2011). The LAI is averaged the model $2° \times 2°$ grid resolution and for each grid cell it is split between PFTs using the PFT grid cell fractional coverage. Photosynthesis and fluorescence are simulated at an hourly time step but forced by the respective

monthly mean diurnal cycle such that a single diurnal cycle simulated for each month.

## 2.6 Global GPP Products for Comparison

To assess the SIF-optimized global GPP we compare the BETHY-SCOPE prior and posterior GPP to other global GPP products. The first dataset for comparison is an upscaled product based on site level measurements termed FLUXCOM GPP (Tramontana et al., 2016). The FLUXCOM GPP product uses machine learning techniques to empirically upscale flux tower data

using remotely sensed data as the predictor variables. The second dataset for comparison is an ensemble of eleven global dynamic vegetation models forced with equivalent climate fields and atmospheric $CO_2$ concentration that were used to investigate trends in sources and sinks of $CO_2$ (TRENDY; Sitch et al., 2015). These GPP estimates are based on their own model assumptions and/or sparse measured data (Anav et al., 2015). They are therefore used to evaluate whether the SIF assimilation results in global patterns of GPP that align with the current understanding and not strictly for validation purposes.

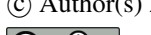



## 3 Results

There is a wide array of results that may be presented from a global data assimilation study using a novel observation such
as SIF. First we present the fit of the prior model and posterior model to the observational data. Following this we examine
estimated parameters and their associated prior and posterior uncertainties. We then show results of spatiotemporal patterns
of optimized GPP alongside GPP estimates from other studies. Finally, we present derived simulated quantities including the
light-use efficiency of GPP and APAR.

### 3.1   Assimilation with SIF

Here we show how the prior and posterior model SIF compare with observed SIF for the calibration and validation periods. We
assess the goodness of fit between model and observed SIF using multiple metrics. The $\chi_r^2$ fit is a key metric as outlined in the
methods. Differences between the model and observations ('residual') and the squared residual normalized by the observational
variance ('mismatch') are also shown. The mismatch provides a measure of the difference between the model and observations
accounting for observational uncertainties, indicating the contribution grid cells make toward the cost function.

We present the model fit over the calibration and validation periods. The model fit during the calibration period is presented
in more detail as there is more data. The model fit during the validation period provides a more stringent test of the assimilation
performance. We then show how the additional model simulation testing seasonal variation in parameters.

### 3.1.1   Calibration

The mapped annual mean OCO-2 SIF for 2015 is shown in Fig. 1. Model-observed residuals in annual mean SIF are shown for
SIF$_{prior}$ and SIF$_{post}$ in Figs. 2 and 3, respectively. Average SIF and the sum of the mismatch is shown per latitude at annual (Fig.
4) and northern summer (June-August; Fig. 5) time scales. Note that because observations do not occur in every spatiotemporal
grid cell, only grid cells with observations are used when calculating the latitudinal averaged SIF. A scatter plot and histogram
of residuals of the modelled and observed SIF for the prior and posterior cases are shown in the Supplement Figs. 3-6. Mapped
model mismatch is shown in the Supplement Figs. 7 and 8.

SIF$_{prior}$ over the calibration period yields a global $\chi_r^2$ fit of 2.24. Large residuals in the annual mean are evident across the
globe, ranging from -0.96 to +0.75 $\mathrm{Wm^{-2}\mu m^{-1}sr^{-1}}$. Generally, SIF$_{prior}$ overestimates observed SIF across regions dominated
by tropical forest (e.g. the Amazon, western equatorial Africa and Maritime Continent), boreal forest (parts of North America
and Eurasia), and arid regions (e.g. central Australia, central Asia and southern Africa). SIF$_{prior}$ tends to underestimates
observed SIF across the rest of the land, in particular for regions dominated by croplands (e.g. central USA, parts of Europe,
India, eastern Asia), mixed forests (across Europe and Asia), and grassland and savanna regions (e.g. African savanna, south-
east South America). Latitudinal averages are in line with these spatial patterns for SIF$_{prior}$. Overestimation of observed SIF is
seen over the central tropics between 15°S and 5°N, a region dominated by tropical evergreen forest (TrEv), whereas there is
significant underestimation of observed SIF over the northern hemisphere, particularly during northern summer (Fig. 5).





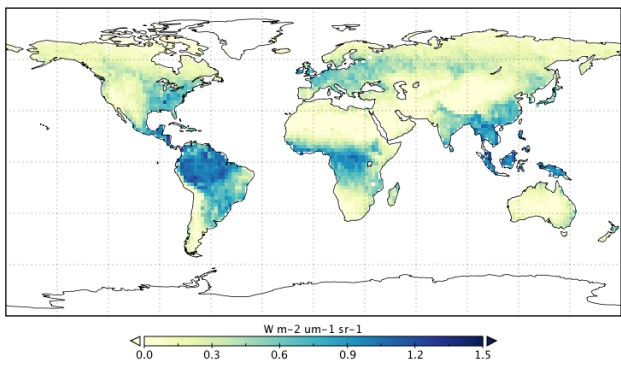

**Figure 1.** Annual mean observed SIF from the OCO-2 satellite for 2015.

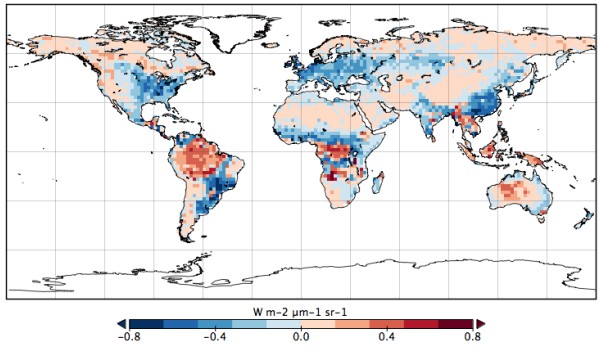

**Figure 2.** Annual mean residual between model SIF$_{prior}$ and observed SIF for 2015.

Following the assimilation, the model shows a considerably better fit to the data. The global $\chi_r^2$ fit is strongly reduced from 2.24 to 1.17, close to the optimal value of one, demonstrating the ability of the optimized model to fit the observed patterns of SIF. Annual mean residuals between SIF$_{post}$ and the observations range between -0.66 and +0.39 $\mathrm{Wm^{-2}\mu m^{-1}sr^{-1}}$, considerably smaller than SIF$_{prior}$ as can be seen in the spatial patterns across the globe (Fig. 2).

5    Latitudinal sums of the mismatch between the model and the observations shown in Figs. 4 and 5, show a significant reduction following the assimilation. There is considerable reduction in mismatch (i.e. improvement in fit) across all latitudes at the annual and northern summer time scales. The total annual mismatch between SIF$_{post}$ and the observations is about 40-60% smaller across the latitudes between 40°S to 60°N relative to SIF$_{prior}$.

Despite the strong improvement in fit, SIF$_{post}$ tends to underestimate large observed SIF values (>1.0 $\mathrm{Wm^{-2}\mu m^{-1}sr^{-1}}$).

10   Compared with these observations only, SIF$_{prior}$ has a poor $\chi_r^2$ fit of 8.27. For SIF$_{post}$ there is an improved $\chi_r^2$ fit of 5.60 but





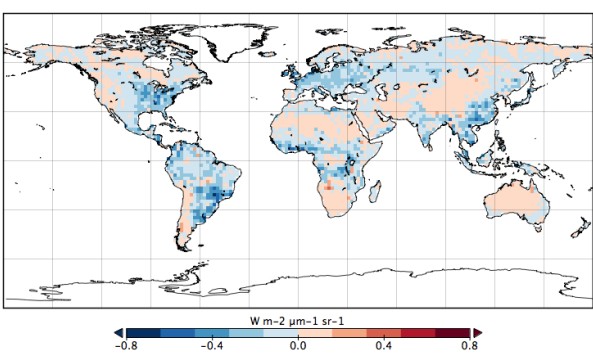

**Figure 3.** Annual mean residual between model SIF$_{post}$ and observed SIF for 2015.

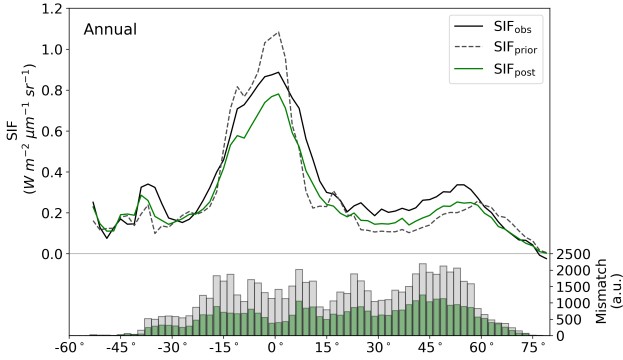

**Figure 4.** Latitudinal averaged SIF and mismatch with observations. OCO-2 observations (black line), SIF$_{prior}$ (grey dashed line, grey bars), and SIF$_{post}$ (green line, green bars) for the annual average. Data is only shown for spatiotemporal points where OCO-2 observations are present.

systematically underestimates the observations by 0.48 $\mathrm{Wm}^{-2}\mathrm{\mu m}^{-1}\mathrm{sr}^{-1}$. These large observed SIF values occur mostly over the northern mid-latitudes during the peak growing season and over the tropics.

From Fig. 3 it appears that SIF$_{post}$ overestimates observed SIF over arid regions (e.g. central Australia, southern Africa, central Asia). This is largely because of observed SIF values that are slightly negative, potentially due to measurement noise or issues from the correction of constant error artifacts in the SIF retrieval (Sun et al., 2018). Negative SIF values are still included in the assimilation system. However, they contribute little to the overall mismatch given the uncertainty in the SIF observations (see Supplement Figs. 7 and 8).

Considering the importance of $V_{cmax}$ for modelling GPP we also assess how the optimized $V_{cmax}$ parameters affect the fit to the observations. To do this we perform a simulation using the SIF$_{post}$ parameter set, but adjust the $V_{cmax}$ parameters back to their prior values. The global $\chi_r^2$ fit for this simulation is 1.20, a minor change compared to SIF$_{post}$ ($\chi_r^2$ = 1.17).





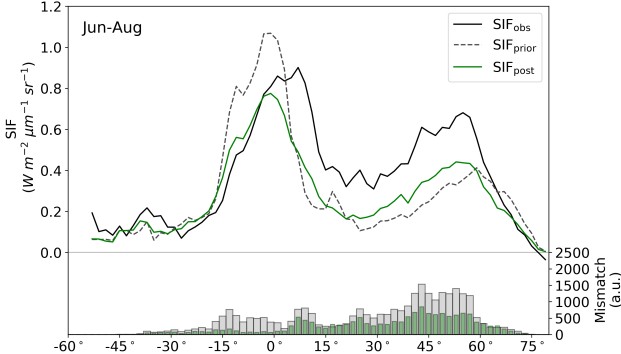

**Figure 5.** Latitudinal averaged SIF and mismatch with observations. OCO-2 observations (black line), SIF$_{prior}$ (grey dashed line, grey bars), and SIF$_{post}$ (green line, green bars) for northern summer (June-August). Data is only shown for spatiotemporal points where OCO-2 observations are present.

### 3.1.2 Validation

To validate the optimized model we assess the model fit to independent OCO-2 SIF data between September-December 2014 which is outside of the calibration period. The LAI is calculated and prescribed to the model using the September-December 2014 data from Yuan et al. (2011). For the prior model SIF (SIF$_{prior}$) the global $\chi_r^2$ fit to this data is 2.10. Following the
assimilation of the 2015 SIF data, the posterior model SIF (SIF$_{post}$) gives a global $\chi_r^2$ fit of 1.04, near the optimal value of one indicating a strong improvement in the fit.

Comparison of the optimized model SIF with the validation data shows that large SIF values (>1.0 $\mathrm{Wm^{-2}\mu m^{-1}sr^{-1}}$) are systematically underestimated. While the observed SIF for any given month and grid cell can reach up to 1.7 $\mathrm{Wm^{-2}\mu m^{-1}sr^{-1}}$, SIF$_{post}$ does not exceed 1.2 $\mathrm{Wm^{-2}\mu m^{-1}sr^{-1}}$. For this validation data, these large SIF values typically occur over tropical
forest, grassland and cropland regions.

### 3.1.3 A Case with Seasonally Varying Parameters

Most terrestrial biosphere models assume process parameters are constant through time despite evidence showing that some key parameters (e.g. $C_{ab}$, $V_{cmax}$) vary in response to resource availability (e.g. Demarez, 1999; Wang et al., 2007; Wilson et al., 2000; Xu and Baldocchi, 2003; Zhang et al., 2014). At present BETHY-SCOPE does not include any mechanism for
varying these with time. Given this, we expect that assuming these parameters are temporally constant will contribute to a large disparity between the model and observations, particularly for more seasonal vegetation.

Thus, an additional comparison is made where we apply seasonal variation to $C_{ab}$ and $V_{cmax}$ parameters for the posterior model. We set the annual mean to be the posterior $C_{ab}$ and $V_{cmax}$ values and apply a seasonal cycle by using a sine function that has a period of one year, a maximum on the summer solstice (i.e. December 22nd in southern hemisphere and June 22nd
in northern hemisphere) and an assigned amplitude. For highly seasonal PFTs including deciduous trees and shrubs, C3 and





C4 grasses, and crops, the amplitude is set to 50% of the mean, while for all other PFTs the amplitude is set to 10%. This provides a simple sensitivity test to investigate whether introducing seasonal variation in $C_{ab}$ and $V_{cmax}$ improves the fit with the observed SIF over the calibration period.

Implementation of seasonally varying $C_{ab}$ and $V_{cmax}$ results in a moderate improvement in fit with the observed SIF.

The posterior $\chi_r^2$ fit improves from 1.17 to 1.10 given the seasonally variable parameters (SIF$_{post,seas}$). Additionally, a linear regression line between observed and modelled SIF shifts slightly closer to the 1:1 line, with the slope of the line increasing from 0.28 to 0.30. This indicates that the systematic underestimation of large observed SIF values ($>1.0$ Wm$^{-2}$μm$^{-1}$sr$^{-1}$) may be improved. Indeed, for these large observed SIF values the $\chi_r^2$ fit (calibration period) improves more substantially than the global fit, from 5.60 for SIF$_{post}$ to 5.17 for SIF$_{post,seas}$. The fit to low SIF values ($<0.25$ W m-2 um-1 sr-1) remains the

same in both cases. When the fit to observed SIF is assessed per PFT (i.e. grid cells with the same spatially dominant PFT are considered together) the $\chi_r^2$ fit improves most significantly for tropical deciduous, temperate deciduous, croplands, and C3 and C4 grasslands.

### 3.1.4  Fit to the Seasonal Cycle

We can also assess the seasonal cycle of SIF to determine how well the model simulates the amplitude of observed SIF. First,

we assess how well model replicates the seasonal amplitude of observed SIF across all spatial points. Second, we assess the seasonal patterns of SIF for a selection of case study regions in more detail. We avoid assessing the seasonal cycle of SIF aggregated at global or hemispheric scales as regional patterns of residuals can differ in sign and magnitude (e.g. see Fig. 3). The seasonal amplitude of observed SIF is calculated as the difference between the maximum and minimum across the year for each grid point. To increase confidence that the observations really capture the seasonal cycle of each grid point, we only

assess grid points with at least eight months of observed SIF data. In doing so, most regions north of 60°N are excluded due to limited sunlight. We do not assess the timing of the seasonal cycle (e.g. start and end of the growing season) considering this is largely driven by LAI which is prescribed and therefore fixed in this study.

The comparison of the seasonal amplitude of observed SIF against SIF$_{prior}$, SIF$_{post}$ and SIF$_{post,seas}$ is shown in the Supplement Fig. 9. The model underestimates the observed seasonal amplitude in all cases. With a perfect match to the observed seasonal

amplitude the model would follow the 1:1 line and have an average ratio of one. However, we find that the average ratio is 0.42 for SIF$_{prior}$, 0.41 for SIF$_{post}$, and 0.44 for SIF$_{post,seas}$. Spatial points with the largest seasonal variations in observed SIF also exhibit the largest model-observed mismatch (Fig. 6).

For more detailed assessment of seasonal patterns we investigate three case study regions: (i) the tropical forest of mainland south-east Asia; (ii) croplands in North America, and; (iii) the north African savanna (see Supplement Figs. 10-15 for details).

These regions are selected as they represent quite different biome types, exhibit varied SIF patterns, and have relatively large posterior model-observed mismatch.

The tropical evergreen forest of mainland south-east Asia exhibits a clear seasonal cycle in observed SIF. The monthly mean observed SIF over this region varies from a minimum of ~0.6 Wm$^{-2}$μm$^{-1}$sr$^{-1}$ in March to a maximum of ~1.4 Wm$^{-2}$μm$^{-1}$sr$^{-1}$ in August (see Supplement Fig. 10). Both SIF$_{prior}$ and SIF$_{post}$ exhibit very different seasonal cycles to the




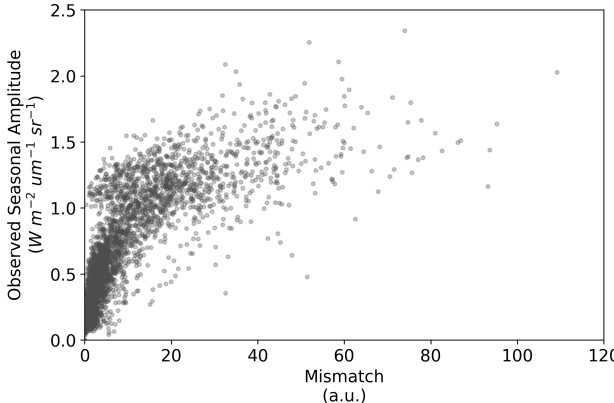

**Figure 6.** Seasonal amplitude of observed SIF versus the model-observed mismatch for the SIF optimized (SIF$_{post}$) model. Mismatch is defined as the squared residual normalized by the observational uncertainty (expressed as variances). A positive curvi-linear relationship exists such that spatiotemporal points with a large model-observed mismatch generally also exhibit a large observed seasonal amplitude.

observations however, with little change in the shape of the seasonal cycle from the prior to posterior simulations. Model SIF shows a minimum in July and two maximums in February and November, following the seasonal evolution of LAI (Supplement Fig. 10). This results in strong negative temporal correlations between observed SIF and model SIF as over this region.

Croplands in North America are heavily managed landscapes with highly productive vegetation as indicated by the large observed SIF values during the growing season. Even with the monthly averages used here, observed SIF can exceed 2.0 $\text{Wm}^{-2}\mu\text{m}^{-1}\text{sr}^{-1}$. As presented earlier, the model SIF cannot match the seasonal amplitude of observed SIF and subsequently underestimates the maximum monthly average SIF over this region by about 35% (0.45 $\text{Wm}^{-2}\mu\text{m}^{-1}\text{sr}^{-1}$) (Supplement Fig. 12). The fit is improved in SIF$_{post,seas}$ ($\chi_r^2$ (SIF$_{post}$) = 1.93; $\chi_r^2$ (SIF$_{post,seas}$) = 1.62). In both cases the timing of seasonal maximum and senescence is simulated quite well, while the onset of the growing season is predicted to be too early.

The north African savanna exhibits a strong seasonal cycle in observed SIF. This region is dominated by grasslands and open forest, with a seasonality closely following the seasonal variation in precipitation. Averaged over the region, observed SIF varies from 0.07 $\text{Wm}^{-2}\mu\text{m}^{-1}\text{sr}^{-1}$ in January to 0.77 $\text{Wm}^{-2}\mu\text{m}^{-1}\text{sr}^{-1}$ in September (Supplement Fig. 14). However, SIF$_{post}$ exhibits little seasonality with variation from 0.22 to 0.41 across the year, only 25% of the observed seasonal amplitude. Temporal correlations are quite strong however, as model SIF also reaching its peak in September.

## 3.2   Estimated Parameters

The prior and optimized parameter mean values and associated uncertainties are shown in Table A1. In this data assimilation system the number of observations far outweighs the number of unknowns. sensitivity This means that there is a substantial amount of observational information available to constrain parameter values, thus they can shift from their prior values considerably even if given a relatively tight prior uncertainty. We can be more confident in parameters that see large reductions in uncertainty. Conversely, parameters with little reduction in uncertainty following optimization should be accepted cautiously.





We focus on two key parameters, $V_{cmax}$ and $C_{ab}$. Additionally, we discuss results from parameters that show either a large change from their prior value or large reduction in uncertainty.

Posterior $V_{cmax}$ estimates range from 11 to 125 $\mathrm{\mu mol m^{-2} s^{-1}}$. The lowest rates are for the C4Gr, Tund, and Wetl PFTs. The highest rates are for the Crop and DecShr PFTs which both exceed 100 $\mathrm{\mu mol m^{-2} s^{-1}}$. Nine out of thirteen PFTs see $V_{cmax}$

increase. Strong increases greater than two standard deviations occur for the PFTs TmpDec, EvCn, C3Gr, and C4Gr. Minor changes of less then half a standard deviation occur for the PFTs TrEv, TmpEv, EvShr, Tund, Wetl, and Crop. Although these estimates are within a reasonable physical range the uncertainty reduction is small (<10%) indicating little constraint from SIF.

Posterior $C_{ab}$ estimates range from 1.3 to 13.1 $\mathrm{\mu g cm^{-2}}$. For all but one PFT the assimilation lowered the prior estimates. The largest posterior $C_{ab}$ values are for Crop, C3Gr, and C4Gr while the lowest are for EvShr and DecShr. Uncertainty reduction

is large for $C_{ab}$ PFTs, all exceeding about 70%, indicating strong constraint by the SIF data. The leaf composition parameter for dry matter content, $C_{dm}$, reduces effectively to zero and sees a 50% uncertainty reduction. The leaf composition parameter for senescent matter fraction remains unchanged at zero and shows only a minor uncertainty reduction of 3%.

Parameters that control canopy structure and the leaf angle distribution see large deviations from their prior values. Some leaf angle distribution parameters, LIDFa and LIDFb, shift considerably. SIF is particularly sensitive to some of the LIDFa

and LIDFb parameters with uncertainty reductions of up to 25%, depending on which PFTs they pertain to. Vegetation height for grasses and crops sees a decrease from 0.5 m to 0.2 m and an uncertainty reduction of 22%. Despite the changes GPP is relatively insensitive to these parameters.

## 3.3   Estimated GPP

In this section we present the effect of the SIF assimilation on the spatiotemporal patterns of model GPP. Following the

assimilation of satellite SIF data global GPP increases by 8.6 $\mathrm{PgC yr^{-1}}$ from 128.4 $\mathrm{PgC yr^{-1}}$ to 137.0 $\mathrm{PgC yr^{-1}}$. This change shifts BETHY-SCOPE closer to the TRENDY mean (142.4 $\mathrm{PgC yr^{-1}}$) but further from the FLUXCOM GPP (103.3 $\mathrm{PgC yr^{-1}}$). The parametric uncertainty in global GPP is reduced by 65% from $\pm 16.6$ $\mathrm{PgC yr^{-1}}$ to $\pm 5.9$ $\mathrm{PgC yr^{-1}}$ following the SIF assimilation.

The spatial patterns of posterior GPP and the changes following the SIF assimilation are shown in Figs. 7-12. A moderate

increase in annual GPP is seen across much of Eurasia and North America (Figs. 8 and 9), mainly due to an increase in temperate forest GPP (PFTs TmpDec, EvCn, DecCn) as shown in Fig. 10. A substantial increase in GPP is also seen for savanna biomes, dominated by C3 and C4 grasses, including over parts of Africa, central Asia, the Brazilian Highlands and parts of Australia. A moderate decline in annual GPP occurs for tundra ecosystems and dry tropical forest regions in South America, Africa, and south-east Asia and parts of South America (dominated by the PFT TrDec) (see Figs. 9 and 10).

Averaged over latitudinal bands (Figs. 11 and 12) the central tropics (15°S-5°N; dominated by the PFT TrEv) show little overall change between the prior and posterior simulations. Prior and posterior estimates are near the high end of other estimates for this region however, larger than the FLUXCOM GPP and TRENDY model average. We note that FLUXCOM GPP is not representative of this region given the sparsity of the flux tower network in the tropics (Tramontana et al., 2016). The northern extratropics show a general increase in GPP however, shifting the BETHY-SCOPE GPP to the higher end of other estimates.





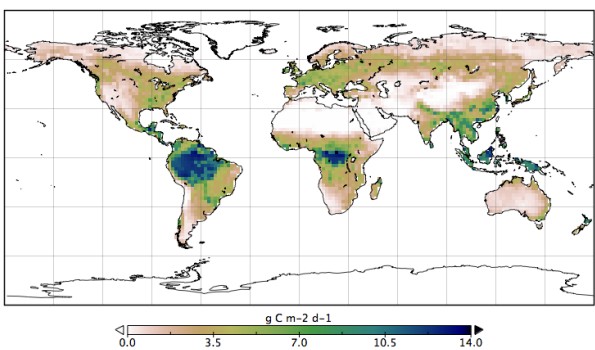

**Figure 7.** Spatial patterns of BETHY-SCOPE posterior annual mean GPP (GPP$_{post}$) for 2015.

GPP over the southern latitudes (south of 15°S) are generally lower than other estimates, but with the assimilation the region south of 30°S shifts closer to both TRENDY and FLUXCOM estimates.

A useful metric for the patterns of global productivity is the ratio of GPP between different regions. The ratio of the tropical (30°S-30°N) to extratropical (south of 30°S and north of 30°N) regions declines following the SIF assimilation, due to an

increase in extratropical GPP while tropical GPP remains relatively unchanged (see Table B1). This shifts the ratio of tropical:extratropical GPP from a prior of 2.59 to a posterior of 2.10, which is substantially closer to patterns of the FLUXCOM (1.91) and TRENDY mean (1.93). Similarly, the ratio of the tropics to the boreal region (north of 55°N), tropics to the temperate region (south of 30°S and north of 30°-55°N), and temperate to boreal region converges closer to FLUXCOM values (see Table B1).

We also note an improvement in the correlation between the BETHY-SCOPE estimate and the FLUXCOM GPP over North America, a region where FLUXCOM GPP has considerably more training data and thus where we expect it to better represent actual GPP. The correlation improves from a prior $R^2$=0.80 to a posterior of $R^2$=0.86 (see Appendix Figs. B5 and B6). Despite this improvement in match between the patterns, the posterior slope is 1.3 indicating that the magnitude of posterior monthly GPP is larger than that of FLUXCOM GPP. A similar change is seen for Europe (data not shown), another region with many

flux tower sites.

Changes in GPP, as caused by changes in parameter values, can be broken down into changes in intercepted radiation (absorbed photosynthetically active radiation; APAR) and photosynthetic light-use efficiency (LUE$_P$). The LUE$_P$ is calculated as the annual average ratio of monthly GPP to monthly APAR. Overall, there are somewhat opposing effects of APAR and LUE$_P$ on GPP from the assimilation of SIF. Globally, there is an increase in LUE$_P$ (Fig. 13 and Appendix Fig. B2) and a

decline in APAR (see Appendix Fig. B4). The decline in APAR is largely due to the decline in $C_{ab}$ for most PFTs (see Table A1). The changes in LUE$_P$ are primarily controlled by changes in the $V_{cmax}$ and $C_{ab}$ parameters, but also partially influenced by other physiological parameters (see Table A1).





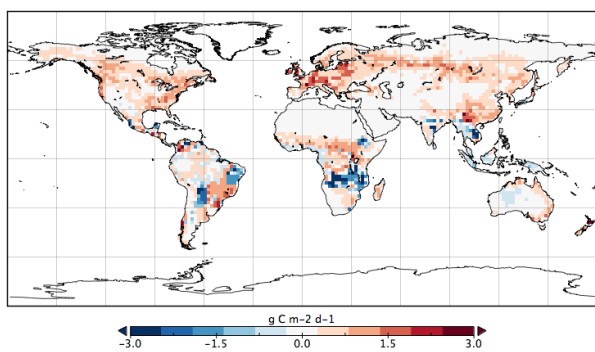

**Figure 8.** Change in annual mean GPP rate for 2015 following optimization with SIF relative to GPP$_{prior}$.

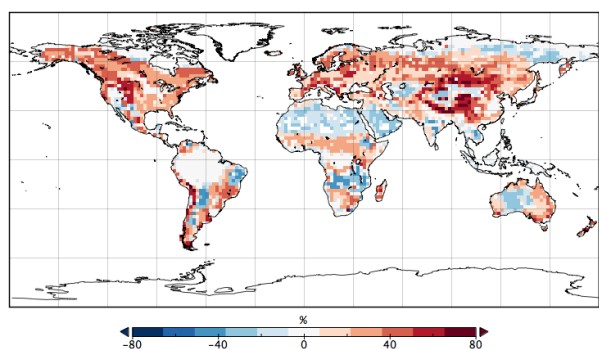

**Figure 9.** Percentage change in annual mean GPP rate for 2015 following optimization with SIF relative to GPP$_{prior}$.

## 4 Discussion

The use of satellite SIF in a data assimilation system has substantially improved the performance of the BETHY-SCOPE model against an independent set of satellite SIF observations. The posterior model fit is slightly better during the validation period ($\chi_r^2 = 1.04$) compared to the calibration period ($\chi_r^2 = 1.17$), indicating that the model performs better outside of the assimilation period. We highlight that this improvement occurs given equivalent LAI fields. Assessing the optimized model in this way is a key validation test and highlights the improvement following the assimilation. While this is the most stringent validation we can carry out with the available data (considering the currently available OCO-2 and climate forcing data), future work should consider longer periods to sample more varied climate forcing conditions. Assessment against other satellite SIF products (e.g. GOME-2, GOSAT) is also feasible provided that careful consideration is taken of the instrumental differences.

With the SIF-optimized model we estimate a global GPP for 2015 of 137.0 $\mathrm{PgCyr}^{-1}$. This is an increase of 7% relative to the prior and is largely due to an increase of GPP in extratropical regions dominated by temperate forests and grassland




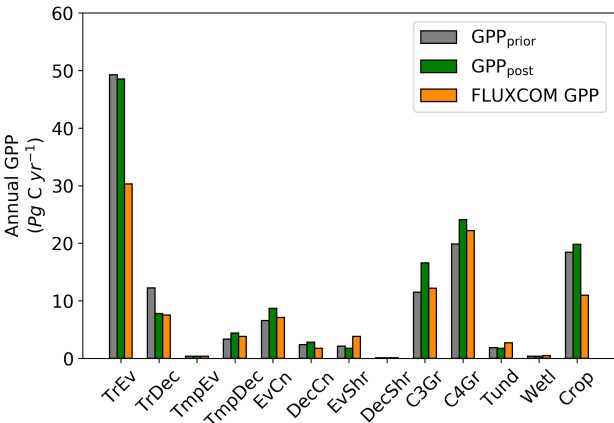

**Figure 10.** Annual GPP for 2015 per biome. Biomes are defined by aggregating model grid cells that have the same spatially dominant PFT as shown in the Appendix Fig. A1.

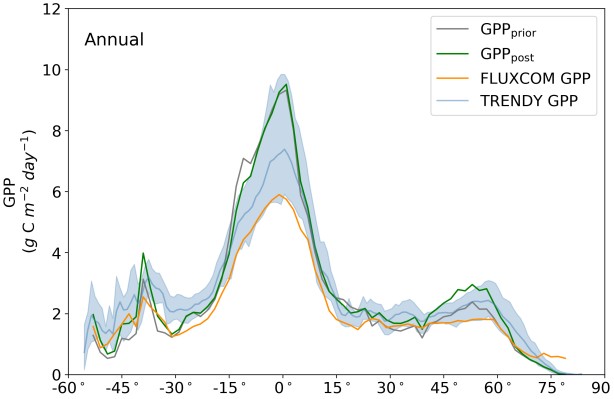

**Figure 11.** Annual latitudinal averages of GPP$_{prior}$ (grey line), GPP$_{post}$ (green line), FLUXCOM GPP (orange line), TRENDY model average (light blue line) and TRENDY model spread given by the 10th and 90th percentiles (light blue shading).

and savanna ecosystems. Other approaches to quantify GPP globally have produced a large range of estimates over different periods including 119 PgCyr$^{-1}$ (Jung et al., 2011), 146 PgCyr$^{-1}$ (Koffi et al., 2012), 157 PgCyr$^{-1}$ (Peylin et al., 2016), and 175 PgCyr$^{-1}$ (Welp et al., 2011). Validating the posterior GPP estimate at these large scales is highly challenging and will require further analysis. The substantial improvement in fit with SIF data during the calibration and validation periods provides some confidence in the overall patterns. We find that the assimilation improves the correlation of BETHY-SCOPE GPP with the FLUXCOM GPP over North America, a region with many calibration sites, but shows a higher magnitude. The SIF assimilation also alters the distribution of global GPP by increasing GPP in extratropical regions. This brings the ratio of the tropics to extratropical regions and the temperate to boreal zone ratio into better agreement with FLUXCOM GPP (Table B1). Previous studies that used SIF to constrain model GPP using linear scaling factors between the two have found similarly




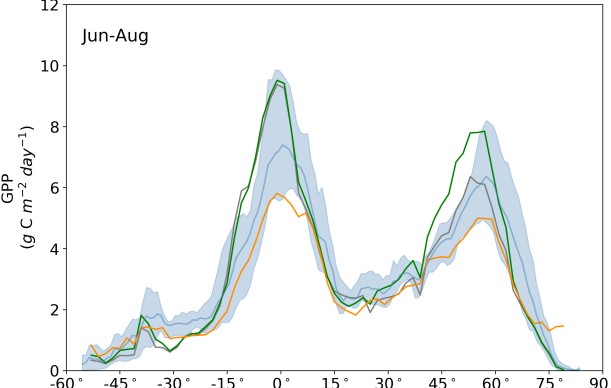

**Figure 12.** Northern summer (June-August) latitudinal averages of GPP$_{prior}$ (grey line), GPP$_{post}$ (green line), FLUXCOM GPP (orange line), TRENDY model average (light blue line) and TRENDY model spread given by the 10th and 90th percentiles (light blue shading).

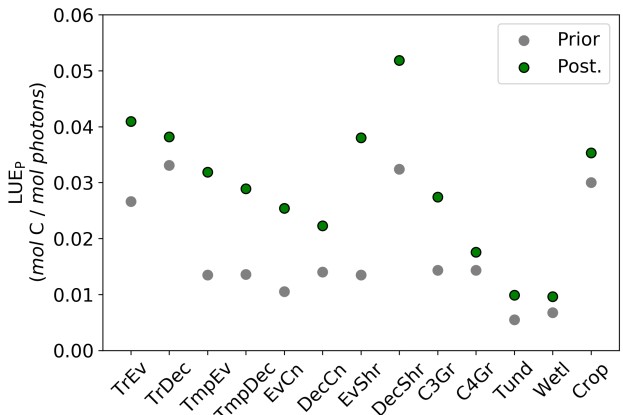

**Figure 13.** Per PFT $LUE_P$ for the prior and posterior simulations. Values were determined by monthly average GPP divided by monthly averaged APAR, then averaged to annual scales thus averaging over seasonal variations. Note that the theoretical maximum is 0.08 molC per molphotons (Waring et al., 2016).

encouraging results (Parazoo et al., 2014; MacBean et al., 2018). In both of these studies an increase in tropical GPP was found, which is in somewhat agreement with our finding of high tropical GPP. Our tropical GPP estimate exceeds both the FLUXCOM GPP and TRENDY model average, suggesting these estimates are too low. Nevertheless, we emphasize that, given the sparsity of the flux tower network, FLUXCOM GPP is not a validation dataset and should be considered with caution particularly over

5    regions with few sites such as the tropics

The uncertainty reduction is large for leaf composition parameters, moderate for canopy structure parameters, and relatively small for leaf physiological parameters. The SIF-constraint on these parameter uncertainties results in a strong overall reduction of parametric uncertainty in global annual GPP of 65%. This differs from previous work that found a global GPP uncertainty




reduction of 73% using the same model (Norton et al., 2018) and a different model (MacBean et al., 2018) which could due to a number of reasons. Firstly, compared with Norton et al. (2018) we use prescribed LAI rather than a prognostic LAI module. Parameters that control LAI were found to be effective at propagating information from SIF to GPP (Norton et al., 2018). The choice to use prescribed rather than prognostic LAI was made due to clear issues with the model simulated LAI,

an issue outside the scope of this study. Secondly, smaller constraint on $V_{cmax}$ by SIF is also found here. In Norton et al. (2018) the constraint is larger due to additional sensitivity of SIF to $V_{cmax}$ via changes in LAI mediated by changes in stomatal conductance, water demand, and a parameter describing the drought sensitivity of LAI (Knorr et al., 2010). In MacBean et al. (2018) a much stronger constraint of $V_{cmax}$ was found as there was no process-based relationship between SIF and GPP such that information is passed directly via linear scaling parameters (i.e. the slope and intercept) to GPP and its related parameters.

The use of linear scaling parameters results in higher parameter error correlations however, putting the posterior $V_{cmax}$ values into question. Finally, compared with Norton et al. (2018), the change in the parameter vector results in different parameter sensitivities (i.e. $H$). Here GPP is more sensitive to $C_{ab}$ compared with Norton et al. (2018) as the assimilation pulls most $C_{ab}$ values into the range where GPP is highly sensitive to them via APAR limitations (see Koffi et al., 2015). This results in strong constraint on GPP via $C_{ab}$ parameters. Overall, these results confirm strong constraint of GPP from satellite SIF data.

The collective change in parameters results in an overall increase of $LUE_P$ and reduction of APAR which has opposing effects on GPP. On it's own, the reduction in APAR would result in reduced GPP. Regions that see a decline in annual GPP (Fig. 8; e.g. dry tropical forests of South America and Africa dominated by the PFT TrDec) show a large decline in APAR and a minor increase in $LUE_P$. The wet tropics (e.g. Amazon) show little change in GPP as there are opposing effects of reduced APAR and increased $LUE_P$. Most other regions see an increase in GPP as the effect of an increased $LUE_P$ outweighs the

effect of reduced APAR. The reduction in APAR is largely driven by the decrease in $C_{ab}$ that occurs for almost all PFTs (Table A1). Posterior values for $LUE_P$ are well within the expected physiological range with the theoretical maximum being 0.08 molCpermolphotons (Waring et al., 2016). Changes in $V_{cmax}$ largely drive changes in $LUE_P$. In general, an increase in $V_{cmax}$ is seen in temperate zone PFTs (TmpDec, EvCn, DecCn, C3Gr, Crop) while a decrease or negligible change is seen in tropical zone PFTs (TrEv, TrDec). The SIF assimilation therefore brings our prior $V_{cmax}$ values into better agreement with

global scale analyses of $V_{cmax}$ that show higher $V_{cmax}$ in temperate zones relative to tropical zones due to higher nitrogen use efficiency (Ali et al., 2015). This presents an opportunity to further evaluate the SIF-optimized global patterns of $LUE_P$ and APAR against independent estimates.

Here, a considerable advance is made on previous studies such that we simulate SIF in a mechanistic way rather than assuming simple linear scaling between SIF and GPP. This allows for the estimation of physically meaningful parameters such

as $C_{ab}$ that may be assessed against in situ or remotely sensed data such as the MERIS Terrestrial Chlorophyll Index. The subtle non-linearity of parameter sensitivities as discussed above cannot be captured with a simple linear scaling between SIF and GPP. This also means we impose knowledge of dynamical limitations on how SIF and GPP relate mechanistically. This includes knowledge of radiative transfer and the non-linear relationship of the quantum yields of PQ, NPQ, and chlorophyll fluorescence.





Deficiencies in the model formulation and/or missing processes still limits the ability of the assimilation system. One example, investigated here, is the lack of time-varying physiological parameters. Introducing time-varying parameters would likely improve the fit to observed SIF and in particular the large SIF values. Introducing an empirical or mechanistic relationship between $C_{ab}$ and photosynthetic capacity (i.e. $V_{cmax}$) via their known relationship to nitrogen content (Evans, 1989) would

also improve the constraint SIF provides on GPP and better represent ecosystem function. SCOPE is a 1D radiative transfer model and therefore may not effectively represent canopies with complex horizontal structure (e.g. open forest). More complex 3D models are under development (Gastellu-Etchegorry et al., 2017) however the high computational requirements will limit their application at the global scale. We note that further work is needed at both leaf and canopy scales to develop the model. The leaf level empirical formulation for NPQ also needs further testing as it partly determines how information is translated

between SIF and GPP via parameters like $V_{cmax}$. Finally, further work is needed to determine a mechanistic basis for drought stress effects on canopy SIF, which should subsequently be implemented in SCOPE.

There are other limitations to this data assimilation. Firstly, it's somewhat limited by use of prescribed LAI. This is exemplified by the regional assessment over the tropical forest of mainland south-east Asia (see Supplement Fig. 10). We point out that the derived MODIS LAI and OCO-2 SIF show different seasonal patterns and that both are uncertain. Nevertheless, with

prescribed LAI the model is limited in its flexibility and cannot alter the shape of the seasonal cycle through the assimilation resulting in a larger posterior mismatch. This may also limit the ability of the model to simulate large SIF values. Secondly, the assimilation algorithm used cannot guarantee the global minimum of $J$ and hence optimal set of parameters, a problem for any local, gradient-based optimization. Thirdly, a number of potential sources of error are not accounted for in the error propagation. This means our uncertainty estimate for global GPP is likely to be an underestimate as it only accounts for uncertainties

from the parameters considered in Table A1. Inclusion of uncertainties in climate forcing and prescribed LAI would increase the uncertainty in global GPP although SIF would mediate this to some extent (Norton et al., 2018). Finally, systematic errors due to the instrument and retrieval errors, spatial sampling biases, and undersampling of diffuse light conditions as thick cloud prevents SIF retrieval may also need addressing in future (Sun et al., 2018). Norton et al. (2018) did note, however, that one of the most important uncertainties arising from the correction of constant error artifacts in the SIF retrieval, did not greatly con-

taminate results. The spatial sampling issues associated with OCO-2 may be overcome with the recently launched TROPOMI instrument that provides daily coverage of the complete Earth.

Future work should assess how SIF and vegetation indices (e.g. EVI, FAPAR) may complement each other in constraining regions of model space, particularly leaf physiological processes. This would require explicit comparisons using the same model. Indeed, knowledge of the mechanistic link of SIF with photosynthetic function suggests that it provides new information

on plant physiological processes that will complement traditional reflectance-based vegetation measurements, as indicated by field studies (Rossini et al., 2015; Walther et al., 2016; Yang et al., 2015). More work is also needed to assess the impact of 3D canopy structure effects on SIF and its relationship with GPP. Further work should also assess the consistency between SIF-optimized GPP and other observational data such as atmospheric $CO_2$, atmospheric carbonyl sulfide, and vegetation indices. These data may be incorporated in a joint assimilation with SIF (e.g. Peylin et al., 2016; Scholze et al., 2016) or used as

independent data for validation purposes.



## 5 Conclusions

In this study we have presented the first application of satellite SIF to optimize parameters of a terrestrial biosphere model with a mechanistic model for SIF. We show, by comparing the model with satellite SIF observations within and outside of the calibration period, that there is substantial improvement in the predictive capability of the model following the optimization
with SIF. Despite this, there are still limitations of BETHY-SCOPE to match the high SIF values. This may be partly due to uncertainties in the prescribed LAI. Varying parameters like $C_{ab}$ and $V_{cmax}$ through time may also improve the match with high SIF values and provide a more accurate representation of ecosystem function. The SIF-optimized GPP is generally higher than the FLUXCOM GPP and TRENDY model average over the central tropics and temperate north. Following the assimilation there is a better match in the spatial contrast of the extra-tropical regions and the tropics compared with FLUXCOM GPP, and a
better correlation with FLUXCOM GPP over North America and Europe that have more flux towers. The use of SIF alters GPP by decreasing APAR and increasing $LUE_P$ across almost all ecosystems. This study provides a significantly useful tool with which to improve our understanding of the global patterns of GPP. This may be extended by applying the model at flux tower sites, using additional satellite SIF data (e.g. GOSAT, GOME-2, TROPOMI), and assimilating other observations relevant to the carbon cycle.

*Code and data availability.*   The BETHY-SCOPE model code is available upon request from the authors. The OCO-2 satellite SIF data is freely available at (insert doi here). Maps were produced using the freely available Panoply software (https://www.giss.nasa.gov/tools/panoply/).




**Appendix A: Spatially Dominant PFT in BETHY-SCOPE Model**

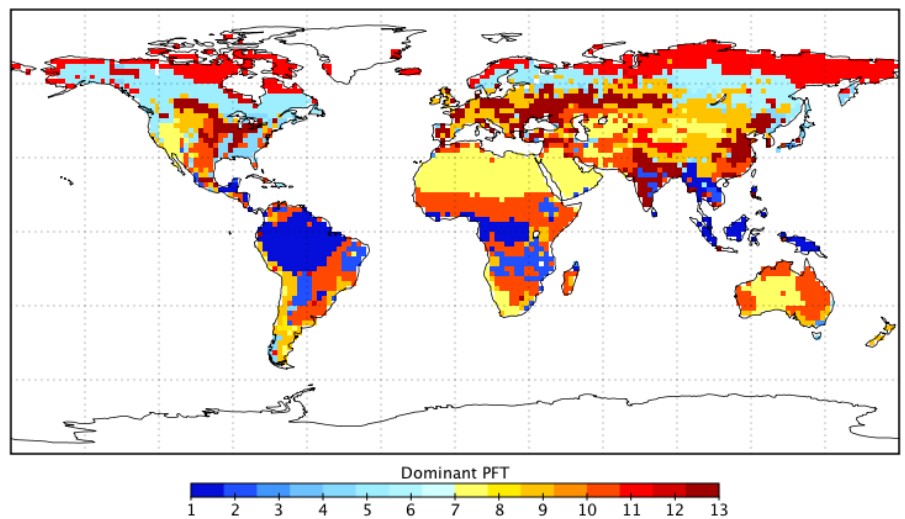

**Figure A1.** Spatially dominant PFT for each BETHY-SCOPE model grid cell.




# Appendix B: Posterior GPP Patterns

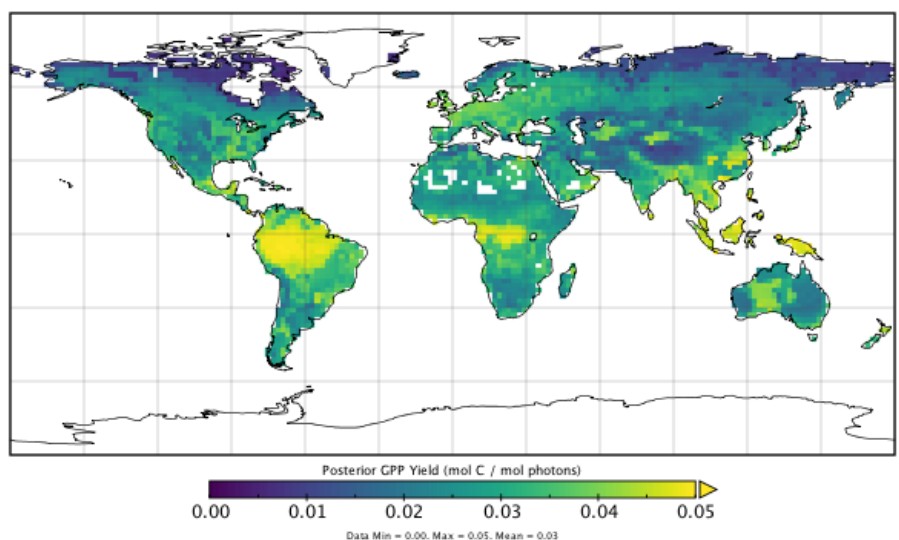

**Figure B1.** Posterior annual mean $LUE_P$ following SIF assimilation.




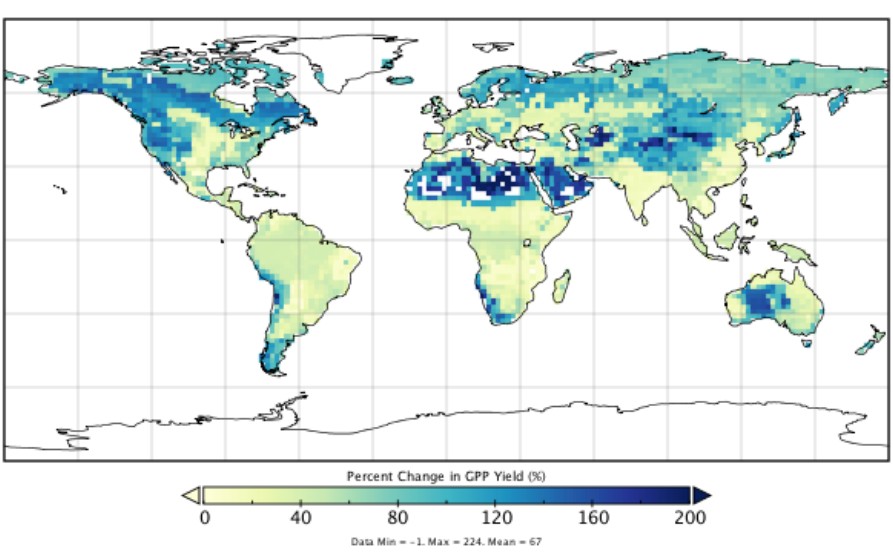

**Figure B2.** Percentage change in annual mean $LUE_P$ following SIF assimilation.




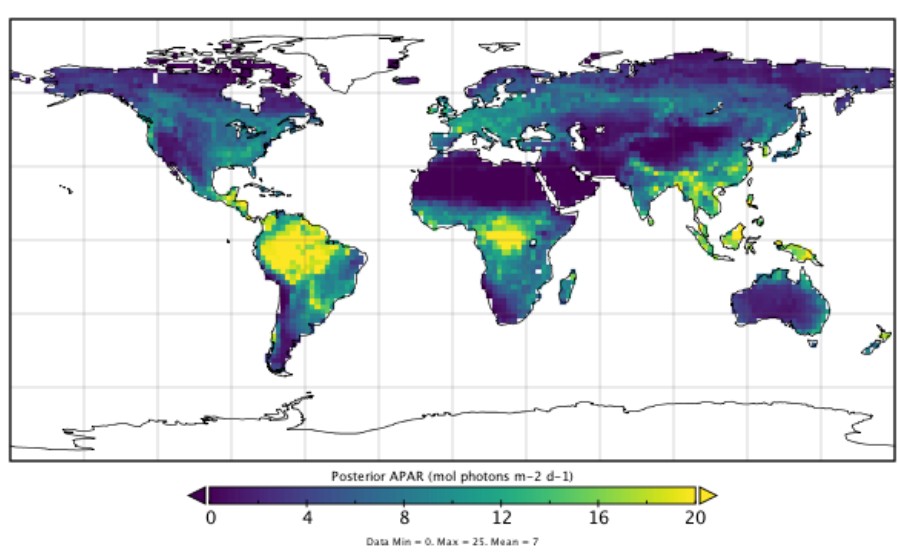

**Figure B3.** Posterior annual mean APAR following SIF assimilation.



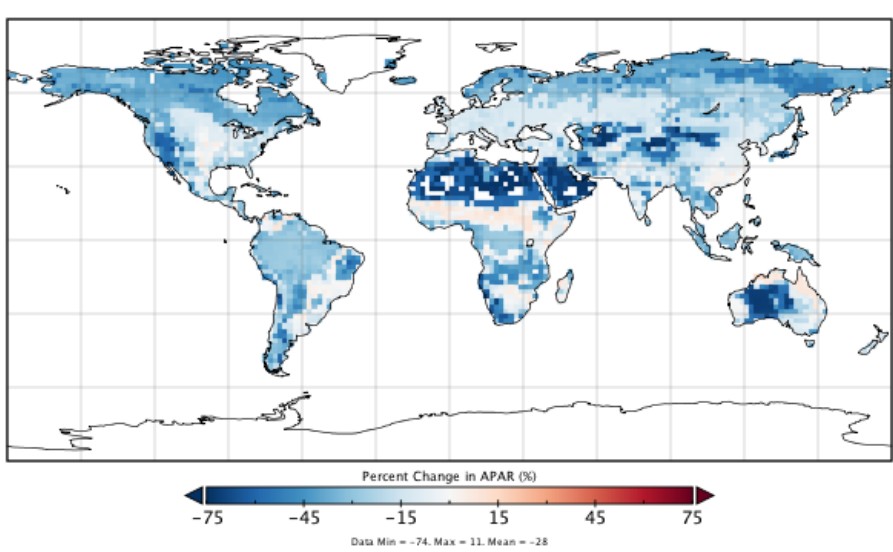

**Figure B4.** Percentage change in annual mean APAR following SIF assimilation.





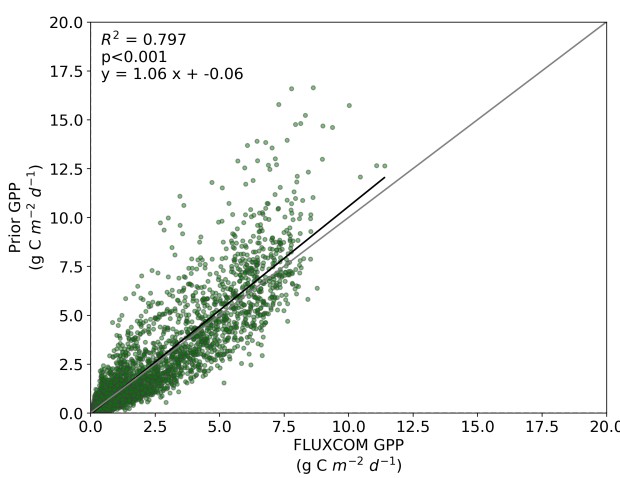

**Figure B5.** Comparison between FLUXCOM GPP and BETHY-SCOPE prior GPP over North America (between 25°N and 55°N).



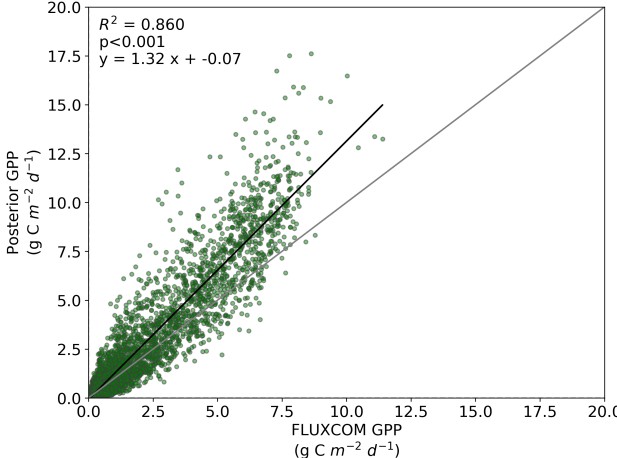

**Figure B6.** Comparison between FLUXCOM GPP and BETHY-SCOPE posterior GPP over North America (between 25°N and 55°N).





**Table A1.** BETHY-SCOPE process parameters along with their prior and optimized uncertainties following SIF constraint, represented as one standard deviation. Relative uncertainty reduction (i.e. effective constraint) is reported for the error propagation with low-resolution and high-resolution SIF observations. Units are: $V_{cmax}$, $\mu$mol($CO_2$) m$^{-2}$ s$^{-1}$; $a_{V_o,V_c}$ and $a_{R_d,V_c}$, dimensionless ratios; $K_C$ and $K_O$, bar; $C_{ab}$, $\mu$g cm$^{-2}$; $C_{dm}$, g cm$^{-2}$; $C_{sm}$, dimensionless fraction; hc, m; leaf width, m. Footnotes: [a] Prior values based on Verhoef and Bach (2007). [b] Applies to PFTs TrEv, TrDec, TmpEv, TmpDec, EvShr, DecShr. [c] Applies to PFTs EvCn, DecCn, Tund. [d] Applies to PFTs C3Gr, C4Gr, Wetl, Crop. [e] Applies to PFTs TrEv, TrDec, TmpEv, TmpDec, EvCn, DecCn. [f] Applies to PFTs. EvShr, DecShr. [g] Applies to PFTs C3Gr, C4Gr, Tund, Wetl, Crop.

| Class | # | Description | Parameter | Prior | Posterior | Change | Effective Constraint (%) |
|---|---|---|---|---|---|---|---|
| LEAF PHYSIOLOGY | 1 | | $V_{cmax}$ (TrEv) | $60 \pm 12$ | $58 \pm 11.2$ | -0.1 | 6.4% |
| | 2 | | $V_{cmax}$ (TrDec) | $90 \pm 18$ | $44 \pm 17.1$ | -2.6 | 5.3% |
| | 3 | | $V_{cmax}$ (TmpEv) | $41 \pm 8.2$ | $41 \pm 8.2$ | 0 | <1% |
| | 4 | | $V_{cmax}$ (TmpDec) | $35 \pm 7$ | $55 \pm 7.0$ | +2.9 | <1% |
| | 5 | | $V_{cmax}$ (EvCn) | $29 \pm 5.8$ | $45 \pm 5.8$ | +2.7 | <1% |
| | 6 | Maximum | $V_{cmax}$ (DecCn) | $53 \pm 10.6$ | $62 \pm 10.6$ | +0.9 | <1% |
| | 7 | carboxylation rate | $V_{cmax}$ (EvShr) | $52 \pm 10.4$ | $54 \pm 10.3$ | +0.2 | <1% |
| | 8 | at 25°C | $V_{cmax}$ (DecShr) | $160 \pm 32$ | $125 \pm 31.9$ | -1.1 | <1% |
| | 9 | | $V_{cmax}$ (C3Gr) | $42 \pm 8.4$ | $78 \pm 8.3$ | +4.3 | 1.1 |
| | 10 | | $V_{cmax}$ (C4Gr) | $8 \pm 1.6$ | $11 \pm 1.6$ | +2.0 | 1.4 |
| | 11 | | $V_{cmax}$ (Tund) | $20 \pm 4$ | $22 \pm 4.0$ | +0.4 | <1% |
| | 12 | | $V_{cmax}$ (Wetl) | $20 \pm 4$ | $21 \pm 4.0$ | +0.2 | <1% |
| | 13 | | $V_{cmax}$ (Crop) | $117 \pm 23.4$ | $124 \pm 22.0$ | +0.3 | 6.0 |
| | 14 | Ratio of $V_{omax}$ to $V_{cmax}$ | $a_{V_o,V_c}$ | $0.22 \pm 0.0022$ | $0.22 \pm 0.0022$ | -0.2 | <1% |
| | 15 | Ratio of $R_d$ to $V_{cmax}$ | $a_{R_d,V_c}$ | $0.015 \pm 0.0015$ | $0.015 \pm 0.0015$ | 0 | 0% |
| | 16 | Michaelis-Menten constant of Rubisco for $CO_2$ | $K_C$ | 350e-6 $\pm$ 17.5e-6 | 322e-6 $\pm$ 17.4e-6 | -1.6 | <1% |
| | 17 | Michaelis-Menten constant of Rubisco for $O_2$ | $K_O$ | $0.45 \pm 0.0225$ | $0.48 \pm 0.0225$ | +1.3 | <1% |





| | | | | | | |
|---|---|---|---|---|---|---|
| **LEAF COMPOSITION** | 18 | | $C_{ab}$ (TrEv) | $40 \pm 10$ | $4.5 \pm 0.4$ | -3.5 | 96% |
| | 19 | | $C_{ab}$ (TrDec) | $15 \pm 7.5$ | $2.5 \pm 0.3$ | -1.7 | 96% |
| | 20 | | $C_{ab}$ (TmpEv) | $15 \pm 7.5$ | $2.1 \pm 0.3$ | -1.7 | 96% |
| | 21 | | $C_{ab}$ (TmpDec) | $10 \pm 5$ | $4.5 \pm 0.4$ | -1.1 | 92% |
| | 22 | | $C_{ab}$ (EvCn) | $10 \pm 5$ | $3.2 \pm 0.5$ | -1.4 | 89% |
| | 23 | | $C_{ab}$ (DecCn) | $10 \pm 5$ | $4.5 \pm 0.9$ | -1.1 | 82% |
| | 24 | Chlorophyll $ab$ content | $C_{ab}$ (EvShr) | $10 \pm 5$ | $1.3 \pm 0.1$ | -1.7 | 97% |
| | 25 | | $C_{ab}$ (DecShr) | $10 \pm 5$ | $1.6 \pm 0.3$ | -1.7 | 93% |
| | 26 | | $C_{ab}$ (C3Gr) | $10 \pm 5$ | $7.7 \pm 1.5$ | -0.5 | 70% |
| | 27 | | $C_{ab}$ (C4Gr) | $5 \pm 5$ | $7.6 \pm 1.4$ | +0.5 | 71% |
| | 28 | | $C_{ab}$ (Tund) | $10 \pm 5$ | $3.7 \pm 0.7$ | -1.3 | 87% |
| | 29 | | $C_{ab}$ (Wetl) | $10 \pm 5$ | $5.6 \pm 1.1$ | -0.9 | 77% |
| | 30 | | $C_{ab}$ (Crop) | $20 \pm 10$ | $13.1 \pm 3.2$ | -0.7 | 68% |
| | 31 | Dry matter content | $C_{dm}$ | $0.012 \pm 0.002$ | $0.000 \pm 0.001$ | -6.0 | 50% |
| | 32 | Senescent material content | $C_{sm}$ | $0 \pm 0.01$ | $0 \pm 0.01$ | 0 | 3% |
| **CANOPY STRUCTURE** | 33 | | $LIDFa^b$ | $0 \pm 0.1$ | $0.32 \pm 0.1$ | +3.2 | 5% |
| | 34 | Leaf inclination | $LIDFa^c$ | $-0.35 \pm 0.1$ | $-0.39 \pm 0.1$ | -0.4 | 5% |
| | 35 | distribution | $LIDFa^d$ | $-1.0 \pm 0.1$ | $-0.77 \pm 0.1$ | +2.3 | 25% |
| | 36 | function | $LIDFb^b$ | $-1.0 \pm 0.1$ | $-0.99 \pm 0.1$ | +0.1 | <1% |
| | 37 | parameters$^a$ | $LIDFb^c$ | $-0.15 \pm 0.1$ | $0.30 \pm 0.1$ | +4.5 | 6% |
| | 38 | | $LIDFb^d$ | $0 \pm 0.1$ | $0.96 \pm 0.1$ | +9.6 | 8% |
| | 39 | | $hc^e$ | $20 \pm 3$ | $18 \pm 3$ | -0.5 | <1% |
| | 40 | Vegetation height | $hc^f$ | $2 \pm 0.4$ | $1.9 \pm 0.4$ | -0.2 | <1% |
| | 41 | | $hc^g$ | $0.5 \pm 0.1$ | $0.2 \pm 0.08$ | -2.8 | 22% |
| | 42 | | leaf width | $0.1 \pm 0.01$ | $0.1 \pm 0.01$ | +1.3 | <1% |



**Table B1.** Estimated GPP per biome and latitudinal region. Biomes are defined by the spatially dominant PFT as shown in Fig. A1. The tropics are defined as the region between 30°S-30°N and the extratropics is as all latitudes outside of the tropics. The boreal region is defined as north of 55°N. The temperate region is defined as south of 30°S and 30°-55°N.

| Biome or Region | Prior GPP (PgCyr$^{-1}$) | Posterior GPP (PgCyr$^{-1}$) | FLUXCOM GPP (PgCyr$^{-1}$) |
|---|---|---|---|
| TrEv | 49.3 | 48.5 | 30.3 |
| TrDec | 12.3 | 7.8 | 7.5 |
| TmpEv | 0.4 | 0.4 | 0.4 |
| TmpDec | 3.3 | 4.4 | 3.8 |
| EvCn | 6.6 | 8.7 | 7.1 |
| DecCn | 2.4 | 2.8 | 1.7 |
| EvShr | 2.1 | 1.8 | 3.8 |
| DecShr | 0.1 | 0.1 | 0.1 |
| C3Gr | 11.5 | 16.6 | 12.2 |
| C4Gr | 19.9 | 24.1 | 22.2 |
| Tund | 1.8 | 1.8 | 2.7 |
| Wetl | 0.3 | 0.4 | 0.5 |
| Crop | 18.5 | 19.8 | 10.9 |
| Tropics | 91.4 (±15.75) | 91.4 (±4.72) | 67.5 |
| Extratropics | 35.3 (±3.40) | 43.5 (±1.99) | 34.0 |
| Boreal | 10.3 (±1.21) | 12.7 (±0.57) | 10.1 |
| Temperate | 26.8 (±2.42) | 32.2 (±1.55) | 25.4 |
| Tropics:Extratropics Ratio | 2.59 | 2.10 | 1.91 |
| Tropics:Boreal Ratio | 8.89 | 7.20 | 6.69 |
| Tropics:Temperate Ratio | 3.42 | 2.84 | 2.66 |
| Temperate:Boreal Ratio | 2.60 | 2.54 | 2.51 |

*Competing interests.* The authors declare that they have no conflict of interest.





*Acknowledgements.* Alexander Norton was partly supported by an Australian Postgraduate Award from the Australian Government and a CSIRO OCE Scholarship. This research benefited from support provided by the ARC Centre of Excellence for Climate System Science (CE110001028). This research was undertaken with the assistance of resources and services from the National Computational Infrastructure (NCI), which is supported by the Australian Government. We acknowledge the efforts of the TRENDY modelling group and thank them for

5   supplying the TRENDY model data. We acknowledge the efforts of the OCO-2 science team and Christian Frankenberg for his assistance with the satellite SIF data.



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
