# Peer review of "Estimating global gross primary productivity using chlorophyll fluorescence and a data assimilation system with the BETHY-SCOPE model"

_Biogeosciences, 2018_

## Referee Comment (RC1) · Anonymous Referee #1 · 12 Jul 2018

GENERAL

This paper tries to improve global GPP simulations by updating model parameters through a data assimilation system. With this system, the model improves simulations of global SIF compared with satellite data. SIF and GPP are jointly connected for plant photosynthesis. The improved SIF-related parameters are expected to improve GPP. Different from previous studies, which assumed linear relations between SIF and GPP, this work connects these two variables through a dynamic vegetation model. Such approach is an important step to assimilate satellite-based SIF data for global

[Figure]

GPP simulations. From this aspect, this paper well fits the scope of Biogeosciences. However, based on the outcome of this work, I think the SIF information is NOT used effectively and the related mechanisms are NOT well explored.

The focus of this work seems to improve SIF simulations, instead of the GPP. The data assimilation system helps reduce SIF biases, mainly by changing Vcmax and Cab, in subtropical and tropical regions. The resultant GPP simulation, however, does not show a reasonable improvement. Instead, global GPP widely increases (Figure 8) especially over subtropical regions where the original model is not bad (Figure 11). The authors claimed that the ratios of subtropics to tropics have improved. However, in my opinion, this is only because the tropical values are not improved while the subtropical ones are worsened. As a result, the major weakness of this work is that the assimilated SIF information is not effective to improve GPP simulations.

Another weakness is that the SIF-GPP relations are not well explained and the causes of changes are not explored. The authors presented a detailed explanation of the data assimilation system. However, the relationship between SIF and GPP is not explained. It is unclear how the GPP and SIF are parameterized in the model and how they are connected. Such missing makes it difficult to understand why Cab and Vcmax show opposite changes for most PFTs, and why LUE and APAR show opposite changes following the posterior parameters. This is the information supposed to make this work outstanding from the previous studies using linear GPP-SIF relations, but the authors failed to present.

SPECIFIC

Figure numbers should be clear. Both figures in main text and supplement share the same numbers. For example, there are two Figure 1 (pages 11 and 35)

Page 6 Line 18: "Close similarities", how to quantify that?

Page 8 Line 23: "we are neither over-fitting or under-fitting the data", how do you know

about this?

Page 9 Lines 10-12: "observations outside of the optimization period from September-December 2014" Generally, the calibration is performed using earlier data and validation is performed in later period.

Page 9 Line 23: "FLUXCOM GPP" the FLUXCOM provides up to six different version of GPP. Which one is used here?

Page 10 Line 31: "significant underestimation of observed SIF over the northern hemisphere, particularly during northern summer" Is this the reason why you try to improve the simulations with 'updated' seasonal cycle?

Page 11 Line 5: how to define and quantify "mismatch"?

Page 13 Line 18: "a sine function". How do you know this sine function works fine everywhere globally? How to determine the peak of this sine function? Do you have any references? In the later analyses, we can seem different seasonal cycles at different locations. For example, Figure S10 shows that observed SIF peaks in August instead of June.

Page 15 Line 17: "the number of unknowns. sensitivity" removing sensitivity.

Page 15 Lines 3-7: Here, changes of Vcmax are derived based on data assimilation system. However, this does not mean the derived parameters are correct. You should compare the parameters with available literatures to know whether these changes are reasonable or acceptable in reality. In addition, the relation between SIF and Vcmax is less clear than photosynthesis. Do GPP and SIF share the same relationships to Vcmax?

Page 16 Line 6: "less then half a standard deviation". Should be "than".

Page 16 Line 24: "The spatial patterns of posterior GPP and the changes following the SIF assimilation" What's the reason for GPP changes? How GPP and SIF are

connected?

Figure 11: The tropical GPP is not improved while the subtropical GPP is worse.

Table A1: How the numbers in the column of 'change' are calculated? For example, Vcmax of TmpDec changes from 35 to 55, but the change is only 2.9. It seems that Cab is reduced by one order for most PFTs. Please find some observations to support the Cab ranges.

---

## Referee Comment (RC2) · Anonymous Referee #2 · 17 Jul 2018

This paper aims at improving the BETHY-SCOPE estimated GPP through assimilating SIF data into the model. The results did show a substantial improvement of simulating SIF over different periods. However, the improvements in GPP are very limited. The presented method of assimilating SIF product in constraining model parameters in estimating SIF&GPP is very interesting and could be potentially used for many other models and also for other RS products. The authors have put extensive focus on describing this data-model assimilation method and related results, but there is generally lacking of information about how the model works, like how these sensitive parameters

influence SIF and GPP and how the SIF-based optimization could potentially improve the GPP estimations. Also, there is a lack of discussion about if the changes of parameters values after optimization make sense.

General comments: In general, there is no clear explanation about how SIF is linked to GPP in the model, which is very central for readers to understand the work. How the parameter Vcmax and Cab is used in the model and how these key parameters regulating the information translated from SIF to GPP are missing. What do these under-estimated high SIF value mean in terms of GPP modelling? I would strongly suggest the authors to add more information in the model description part and a deep discussion about potential linkage of uncertainties from SIF evaluations to GPP estimations.

The authors argue there is an improvement of GPP in global distribution relative to independent estimates after assimilate SIF into the model (from Abstract). However, from the result (Figs. 11 and 12, the absolute value of GPP in Table B1), we can see very limited improvements (sometimes worse estimations), relative to the FLUXCOM and TRENDY products. The authors refer the improvements to the closer value of GPP ratio between tropical and subtropical regions, but this ratio is mainly influenced by the increase of GPP in northern extratropical regions. This increase of GPP in the extratropical region after data-model assimilation is however not closer to the FLUXCOM and TRENDY products.

There are certain/large limitations in terms of satellite-derived LAI for some regions (like tropics). This work use prescribed LAI from MODIS as inputs, so how the limitation in MODIS-derived LAI could potentially contribute to the discrepancies we see between the modelled GPP (both before and fater the data-model assimilation) and the other two products? The authors could potentially test use other ecosystem model-based LAIs to drive the model and figure out the impacts of prescribed LAI on the GPP estimations.

About structure: There are large parts of text describing methods (e.g., in Section 3.1.3

and 3.1.4) were placed in the result section. The authors should re-arrange the text a bit. Also a clear description of different sensitivity tests (like Vcmax, seasonality, etc.) and associated reasons for these tests to fit the aim of this study should be added in the method section.

A lot of discussion text with references has been placed in the result section. The authors would consider merging result and discussion sections.

More detailed comments are listed here:

1. Any special reason to choose 2015 as calibration period, and use a few month data from 2014 as validation data, not opposite?

2. P3, L15: what is observation operator?

3. P4, L1-2: Does the model simulate the fractional coverage? Or the fractional coverage from some data?

4. P5, l14, "... we assign relatively large prior uncertainties..." what are the methods used for defining prior uncertainties?

5. Section 2.2, Strongly suggest to add a column with short explanation of each parameter. It is difficult to read this table alone.

6 Both SIF and LAI data are gridded to 2 by 2 degree resolution, which interpolation method was used? Please mention it.

7. P13, L9-10, "... these large SIF values typically occur over tropical forest, grassland and cropland regions... " can the authors explain it why?

8. P20, Line 2-5, if the FLUXCOM GPP is not a validation data and the values are too low, as mentioned, could the authors elaborate a bit more why they think the FLUX-COME GPP is too low?